# Time-course of host cell transcription during the HTLV-1 transcriptional burst

Helen Kiik[1,☯¤], Saumya Ramanayake[1☯¤], Michi Miura[1¤], Yuetsu Tanaka[2], Anat Melamed[1], Charles R. M. Bangham[1]*

**1** Department of Infectious Diseases, Faculty of Medicine, Imperial College London, London, United Kingdom, **2** Department of Infectious Disease and Immunology, Okinawa-Asia Research Center of Medical Science, Faculty of Medicine, University of the Ryukyus, Nishihara, Okinawa, Japan

☯ These authors contributed equally to this work.
¤ Current address: Department of Microbiology, Kawasaki Medical School, Kurashiki, Okayama, Japan
* c.bangham@imperial.ac.uk

**Data Availability Statement:** The RNA-sequencing data have been deposited in NCBI's Gene Expression Omnibus (Edgar, 2002) with the accession number GSE197110 (https://www.ncbi.

## Abstract

The human T-cell leukemia virus type 1 (HTLV-1) transactivator protein Tax has pleiotropic functions in the host cell affecting cell-cycle regulation, DNA damage response pathways and apoptosis. These actions of Tax have been implicated in the persistence and pathogenesis of HTLV-1-infected cells. It is now known that *tax* expression occurs in transcriptional bursts of the proviral plus-strand, but the effects of the burst on host transcription are not fully understood. We carried out RNA sequencing of two naturally-infected T-cell clones transduced with a Tax-responsive Timer protein, which undergoes a time-dependent shift in fluorescence emission, to study transcriptional changes during successive phases of the HTLV-1 plus-strand burst. We found that the transcriptional regulation of genes involved in the NF-κB pathway, cell-cycle regulation, DNA damage response and apoptosis inhibition were immediate effects accompanying the plus-strand burst, and are limited to the duration of the burst. The results distinguish between the immediate and delayed effects of HTLV-1 reactivation on host transcription, and between clone-specific effects and those observed in both clones. The major transcriptional changes in the infected host T-cells observed here, including NF-κB, are transient, suggesting that these pathways are not persistently activated at high levels in HTLV-1-infected cells. The two clones diverged strongly in their expression of genes regulating the cell cycle. Up-regulation of senescence markers was a delayed effect of the proviral plus-strand burst and the up-regulation of some pro-apoptotic genes outlasted the burst. We found that activation of the aryl hydrocarbon receptor (AhR) pathway enhanced and prolonged the proviral burst, but did not increase the rate of reactivation. Our results also suggest that sustained plus-strand expression is detrimental to the survival of infected cells.

## Author summary

Human T-cell leukemia virus type 1 (HTLV-1) causes a lifelong infection that results in disease in ~10% of cases. The HTLV-1 transactivator protein Tax is involved in both the

nlm.nih.gov/geo/query/acc.cgi?acc=GSE197110).
Scripts used in these analyses are available on
GitHub at https://github.com/
ImperialCollegeLondon/Timerprotein-2022. All
other relevant data are within the Supporting
Information files.

**Funding:** This work was supported by the
Wellcome Trust UK https://wellcome.ac.uk/
(Investigator Award 207477 to CRMB), and the
Medical Research Council UK https://mrc.ukri.org/
(Project Grant MR/K019090/1 to CRMB). The
funders had no role in study design, data collection
and analysis, decision to publish, or preparation of
the manuscript.

**Competing interests:** The authors have declared
that no competing interests exist.

persistence of infected host cells, and the pathogenesis of HTLV-1 infection. *tax* is transcribed from the plus-strand of the provirus, and *tax* expression is not constitutive, but limited to transcriptional bursts. How these bursts affect host cell transcription is not completely understood. Here, we studied the temporal changes in host transcription during successive phases of the plus-strand burst in two naturally-infected T-cell clones. We found that the deregulation of genes involved in Tax-associated processes, including NF-κB activation, cell-cycle regulation, DNA damage response and suppression of apoptosis, coincided with the early phase of the plus-strand burst: these transcriptional effects appear to be limited to the duration of the proviral plus-strand expression. Regulation of cell-cycle genes diverged between the clones, demonstrating the heterogeneity of naturally-infected cells. We observed a pro-apoptotic response, which outlasted the burst and may indicate increased risk of apoptosis following the burst. Finally, we observed that AhR activity regulated the intensity and duration of the burst, but not the dynamics of reactivation.

## Introduction

Human T-cell leukemia virus type I (HTLV-1) is a pathogenic retrovirus that mainly infects CD4$^+$ T-cells, causing a lifelong infection in the host. An estimated 10 million people in the world are living with the virus; between 5% and 10% of infected hosts develop one of the associated diseases Adult T-cell leukemia (ATL) or HTLV-1-associated myelopathy/tropical spastic paraparesis (HAM/TSP) [1,2].

In addition to the canonical retroviral genes, the provirus expresses several regulatory proteins. The sense and antisense strands of the provirus encode the viral transactivator Tax protein and HBZ (HTLV-1 bZIP protein), respectively: these two proteins promote the proliferation and survival of HTLV-1-infected cells, and both have been implicated in the development of ATL [3,4]; however other regulatory proteins may also be involved.

Tax potently modulates proviral and cellular transcription, which has contrasting consequences in stimulating cell cycle progression and proliferation [5–10], or causing temporary cell cycle arrest and senescence [11–14]. HTLV-1-infected or Tax-transduced cells are also protected from apoptosis [15,16]. Transcriptional down-regulation of pro-apoptotic factors [17,18], and up-regulation of anti-apoptotic factors [19–22] are likely to contribute to the protective activity of Tax. By contrast, there is evidence that Tax promotes apoptosis [23–25], and Tax-expressing cells are more susceptible to cell death following exogenous DNA damage [26,27]. Tax expression impairs the functions of p53 [28], causes genome instability, induces double-strand DNA breaks, and inhibits DNA damage response pathways [5,29–32].

HBZ opposes many functions of Tax including proviral transcription, likely mediated by its interactions with the transcription factors CREB, c-JUN and CBP/p300, and by suppression of NF-κB [33–37].

The mechanisms of the pleiotropic effects of HTLV-1 proviral expression remain unclear, and while many important findings have been made with *tax*-transfected cell lines or long-term *in vitro* transformed cell lines, it is a long-standing question how these observations apply to untransformed, naturally-infected T-cells. It is now clear that the *tax* and *HBZ* genes are not constitutively transcribed at the single-cell level in naturally-infected cells *in vivo*, but rather in intermittent bursts [20,38,39], and it is not understood how the diverse observations on cell proliferation and apoptosis are related to these bursts. We studied two naturally-infected CD4$^+$ T-cell clones competent in Tax expression (3.60 and TBX4B), isolated by limiting

dilution from peripheral blood mononuclear cells (PBMCs) of HTLV-1-infected subjects [40], to quantify host and viral transcription during proviral reactivation. Each clone was stably transduced with a reporter construct, under the control of a Tax-responsive promoter that expresses a fluorescent protein–the Timer Protein–which undergoes a time-dependent change in emission frequency. This approach made it possible to separate the plus-strand transcriptional burst into successive phases. Here we report the changes in transcription in the host CD4[+] T cell accompanying the onset and the progression through the HTLV-1 plus-strand burst.

## Results

### I. The Tax-responsive Timer separates temporal phases of spontaneous HTLV-1 proviral reactivation

The Fluorescent Timer protein [41], which changes its emission of blue fluorescence to red fluorescence during chromophore maturation, allows temporal analysis of cellular processes. It has been applied to study the *in vivo* dynamics of both regulatory T-cell differentiation in mice [42], and *Foxp3* expression in inflammation [43].

Two naturally HTLV-1-infected T-cell clones designated TBX4B and 3.60 were stably transduced with a Tax reporter system containing 5 tandem repeats of the Tax-responsive element (TRE) type 2 linked to a truncated HTLV-1 5′LTR (long terminal repeat) and the Timer Protein gene (Fig 1A) [41]. The purified Timer protein initially fluoresces blue, reaching maximum intensity in 0.25 h, and then matures to the red-emitting form with a half-time of 7.1 h, reaching a plateau between 20–25 h (Fig 1B) [41]. The half-life of the blue fluorescence in mouse lymphocytes is ~4 h [42,43].

The HTLV-1 plus-strand burst begins with transcription of *tax*, resulting in a positive-feedback loop of potent activation of plus-strand transcription by Tax protein [44]. Expression of Tax protein is a surrogate for the proviral plus-strand transcriptional burst, and in this study the subsequent induction of the Timer protein by Tax was used to distinguish successive phases of the plus-strand burst. The clones were flow sorted into four populations based on the fluorescence of the Timer protein during spontaneous proviral expression, representing respectively silent proviruses (non-fluorescent) and the early phase (blue), mid-phase (blue-red) and late phase (red) of the plus-strand burst (Fig 1). Following termination of the burst and decay of remaining red fluorescence of the Timer protein, the cells re-enter the silent (non-fluorescent) double-negative (DN) state.

PolyA-selected RNA samples from each clone were sequenced. Clone 3.60 has a 202 bp deletion that lies in the coding region of *env* on the plus-strand and the 3′UTR of *hbz* on the minus-strand (S1 Fig). The deletion did not impair the expression of Tax protein, as shown by the expression of the Timer protein, the up-regulation of both the *Timer* and the plus-strand transcripts of the provirus in the RNA-seq data (Fig 2A). The expression trajectories of both the *HTLV-1 plus-strand* and the Tax-responsive *Timer* transcripts were closely similar in both clones (Fig 2A).

Principal component analysis of the respective Timer phases (Fig 2B) indicated distinct profiles of gene expression during the successive phases of proviral reactivation. Time-series differential expression analysis of the phases of proviral expression identified 10048 significantly differentially expressed (DE) genes in clone 3.60, and 4798 DE genes in clone TBX4B, respectively representing 57% and 29% of expressed host genes (Likelihood-ratio test (LRT); FDR adjusted p-value < 0.01) (S1 and S2 Data). *HTLV-1 plus-strand* and the Tax-responsive *Timer* were among the top most significantly DE genes in each clone (S1 and S2 Data) and *HTLV-1 plus-strand* was the most significantly up-regulated gene in the early burst (Blue)

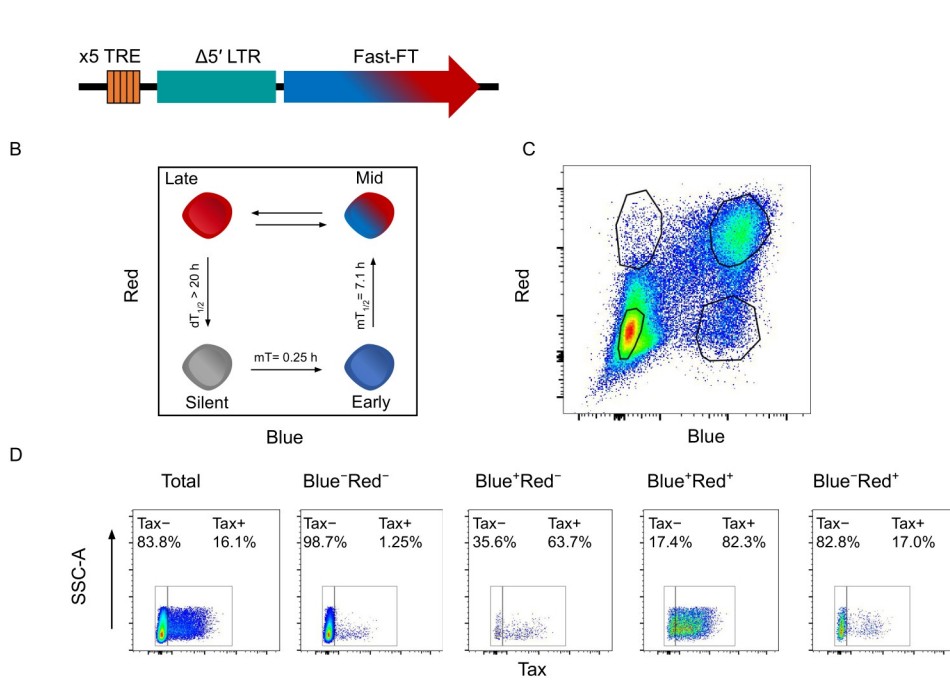

**Fig 1. The principle behind Tax-induced Timer protein expression.** (A) Tax-responsive reporter construct containing the Timer protein gene *Fast-FT*. (B) Schematic of Timer protein expression during progression of the HTLV-1 plus-strand burst. (C) Representative gating strategy used to flow-sort four cell populations for RNA-seq analysis. (D) Tax expression in each respective Timer population, quantified by intracellular staining.

population in each clone, with a log2 fold change (LFC) of 9.60 in clone 3.60 and a LFC of 9.13 in clone TBX4B (Wald test; FDR adjusted p-value < 0.01) (S1 and S2 Data). The *Timer* had a LFC of 6.34 and 5.72 in clones 3.60 and TBX4B, respectively. The second most significantly up-regulated gene during onset of proviral expression in TBX4B was *PNPLA3* (S2 and S3 Figs); the integration site of HTLV-1 in TBX4B lies between exons 2 and 3 of *PNPLA3*.

The trajectory of proviral minus-strand expression had no consistent relationship with plus-strand expression and differed between the two clones (Fig 2A). However, the expression of minus-strand expression in each clone closely resembled that of *SP1*, a known regulator of its transcription [45] (S3 Fig).

Further validating the Timer Protein reporter system used, NF-κB transcription factor genes *REL*, *RELB*, *NFKB1*, *NFKB2* were up-regulated in both clones (Fig 2A). The up-regulation of genes known to be expressed in response to Tax including *IL2RA*, *IL13* and *JUND* [46–49] was confirmed in both clones, and the Tax-repressed target *LCK* [50] was down-regulated (Fig 2A). The expression of *KAT2B* (P/CAF), which interacts with Tax to increase expression from the viral LTR [51], was unexpectedly down-regulated in both clones (S3 Fig).

To examine in detail the transcriptional effects shared by the two clones, the overlap of 3851 genes differentially expressed in both clones was analyzed through K-means clustering. Using k = 5 produced five clusters depicting respectively genes up-regulated during the early burst in each clone, genes up-regulated with a delayed peak (in mid-burst) in each clone, genes down-regulated in each clone, and two clusters showing genes with opposite trajectories (Fig 3A).

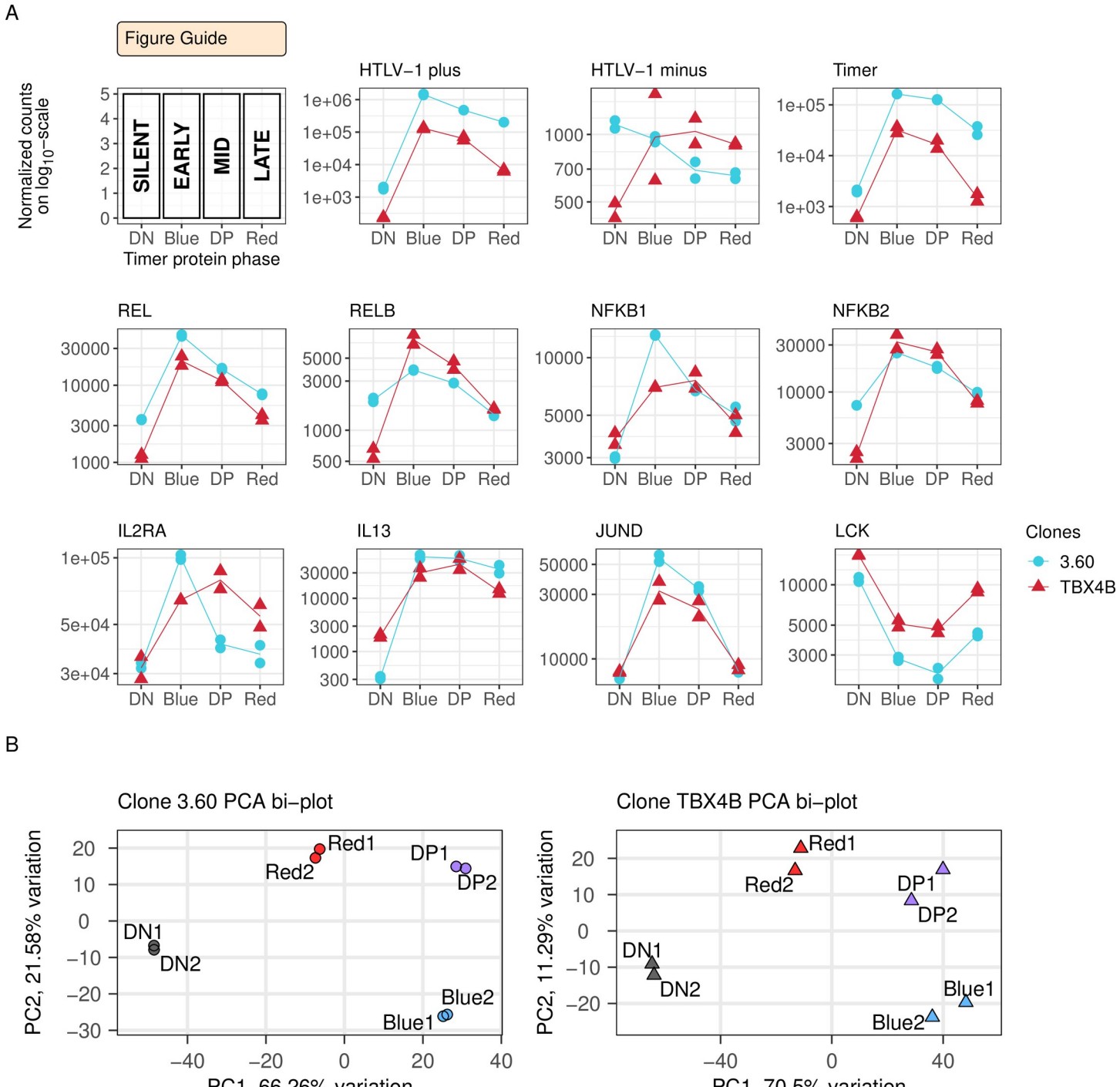

**Fig 2. RNA-seq validates the experimental setup.** (A) Expression of HTLV-1 plus- and minus-strands quantified by RNA-seq in each Timer population. Statistical significance was determined by the likelihood-ratio test (LRT). FDR-corrected p-value < 0.01; ns—not significant. (B) Principal component analysis (PCA) bi-plot of the RNA-seq data in each clone.

To infer functional characteristics of these clusters, an over-representation analysis (ORA) of the MSigDB Hallmark gene set [52] was performed. This analysis identified "TNFα signalling via NF-κB" as the most significant term in two clusters: immediately up-regulated genes

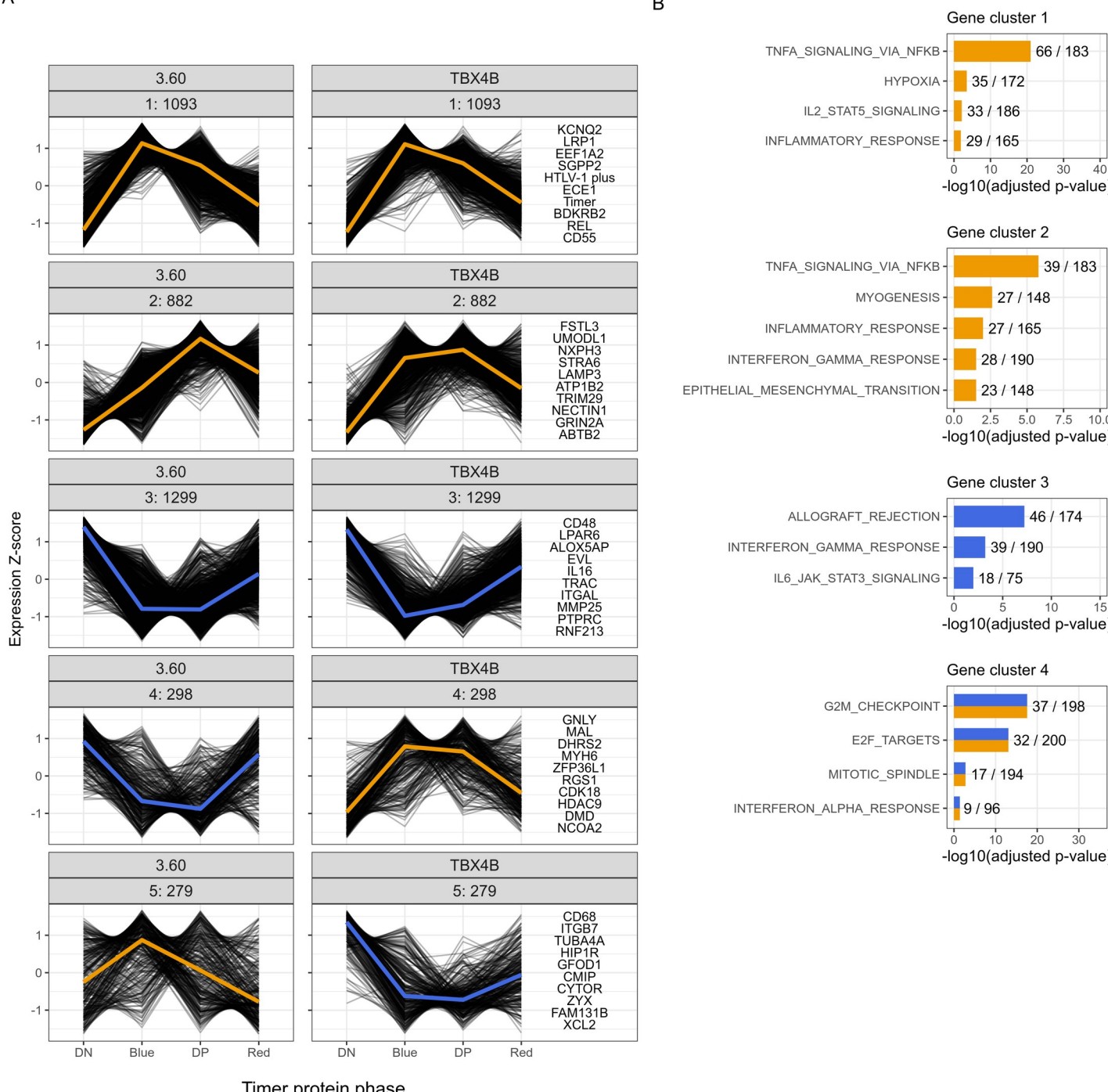

**Fig 3. Genes deregulated in both clones separate into clone-independent and clone-specific clusters.** (A) K-means clustering of 3851 shared significantly DE genes. Statistical significance was determined by LRT. FDR-corrected p-value < 0.01. The top 10 genes in each cluster, based on mean rank of sorted p-values, are listed on the right of the panel. The mean expression trajectory is coloured as a yellow or blue line representing upregulation and downregulation, respectively. (B) ORA of K-means clusters with the Hallmarks gene set from The Molecular Signatures Database (MSigDB). Statistical significance was determined by Fisher's exact test in g:Profiler. FDR-corrected p-value < 0.05.

during plus-strand expression (Fig 3 cluster 1) and genes with a delayed peak of expression during mid-burst (Fig 3 cluster 2). *TNF* itself was downregulated (S1 and S2 Data). "Hypoxia" was similarly an enriched term in the up-regulated cluster 1; *HIF1A* itself was upregulated (S3 Fig). Other enriched terms in cluster 1 were "IL2-STAT5 signaling" and "Inflammatory response", the latter was enriched in cluster 2 as well. "TNFα signalling via NF-κB" and "Hypoxia" were also enriched when all differentially expressed genes in each clone were clustered and analyzed separately (S4B Fig cluster 1 and S4D Fig cluster 1).

Cluster 3 consisted of genes down-regulated in both clones and was enriched for "Allograft rejection", "IL6/JAK/STAT3 signaling" and "Interferon gamma signaling" (Fig 3). "Interferon-γ response" was significant in both the delayed up-regulated gene cluster 2 and the down-regulated gene cluster 3 (Fig 3).

The fourth cluster contained genes that were differentially expressed in both clones, but these genes were down-regulated in clone 3.60 and up-regulated in TBX4B. This cluster included cell-cycle-related genes with Hallmark terms "G2M checkpoint, E2F targets", "Mitotic spindle". Cell-cycle-related genes are analysed in more detail below.

Cluster 5, which represented genes up-regulated in clone 3.60 and down-regulated in clone TBX4B, did not result in any significantly enriched Hallmark terms (Fig 3).

## II. Clone-specific association between proviral expression and cell cycle genes

We investigated how the contrasting observations of Tax induced cell proliferation or cell cycle arrest and senescence relate to the transcriptional control of host genes in naturally infected T-cells during successive phases of proviral reactivation.

For a systematic analysis of genes associated with different cell cycle stages, cyclically expressed genes were obtained from the online database Cyclebase 3.0 (https://cyclebase.org/CyclebaseSearch). All differentially expressed genes from each clone were separated into distinct groups of genes, based on the Cyclebase classification, each group with peak expression in different cell cycle phases (G1, G1/S, G2, G2/M, M). The results (Fig 4A) show that genes associated with peak expression during different phases of cell-cycle progression were down-regulated in clone 3.60 during plus-strand expression, but were up-regulated in TBX4B. These groups contained genes with established roles in DNA replication including *GINS2*, *CHAF1B*, as well as phosphatases *CDC25A*, *CDC25C*, kinases *PLK1*, *AURKB*, *NEK2*, mitotic-spindle-related genes *PRC1*, *BIRC5*, *CDCA8*, and the known marker of cell proliferation *MKI67*. In clone TBX4B, expression of G1-S phase-related genes peaked during the early burst and G2-M phase genes peaked during the mid-burst (Fig 4A).

The trajectories of G1 phase cyclin-dependent kinase *CDK6* and cyclins *CCND1*, *CCND2*, *CCND3*, which mediate entry into the cell cycle, were similar in the two clones (Fig 4B). *CDK6*, but not *CDK4*, was significantly up-regulated in both clones and highest during the early burst (Blue) of proviral plus-strand expression (Figs 4B and S3). Both *CCND1* and *CCND2* were significantly up-regulated in clone 3.60, whereas the expression of *CCND1* was much lower in TBX4B; *CCND2* showed a trend of up-regulation, which was not significant. *CCND3* was significantly down-regulated during the early burst in both clones.

*CDK2* (active in G1/S and S phase) was up-regulated in both clones during early and mid-burst (Fig 4B). By contrast, the G1/S phase cyclins *CCNE1* and *CCNE2* differed between the clones. *CCNE1* was expressed at a low level, and its subsequent up-regulation was delayed until mid-burst in 3.60; its trajectory was inconclusive in TBX4B. By contrast, *CCNE2* was significantly up-regulated only in TBX4B. More importantly, the G1/S phase transcription factor *E2F1* was significantly up-regulated in TBX4B and down-regulated in 3.60.

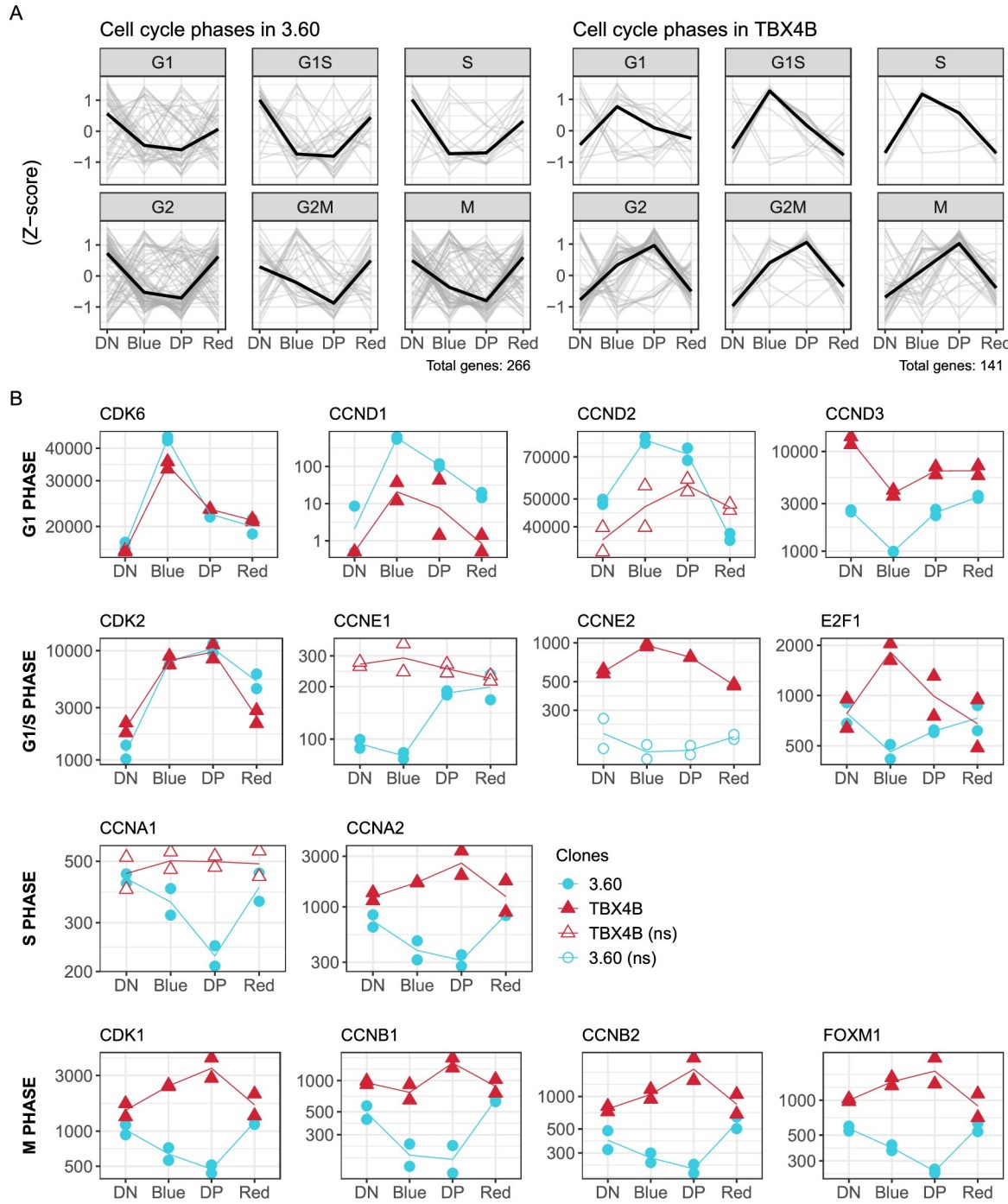

**Fig 4. Divergent association between proviral plus-strand expression and genes related to the cell cycle.** (A) Trajectories in each infected clone of differentially expressed Cyclebase 3.0 genes associated with G1, G1/S, S, G2, G2/M and M phases of the cell cycle. (B) Trajectories of cyclins, CDKs and transcription factors. Y-axis: normalized counts on log$_{10}$-scale. Statistical significance was determined by LRT. FDR-corrected p-value < 0.01; ns—non-significant.

The pattern of cyclin expression progressively diverged between the clones as the cell cycle advanced (Fig 4B). The S-phase genes *CCNA1* and *CCNA2* were down-regulated in clone 3.60, whereas *CCNA2* was up-regulated in TBX4B. Similar contrasts in expression were observed in

mitotic cyclins *CCNB1*, *CCNB2*, *CDK1* kinase, and the mitotic phase transcription factor *FOXM1*. The expression level of many of these genes returned to the value seen in the silent phase by late burst, when proviral expression is terminating. This divergent gene expression of cell cycle regulators between the two clones demonstrates that two naturally infected T-cell clones can fundamentally differ in their response to proviral plus-strand expression.

## III. Immediate up-regulation of genes involved in the DNA damage response is followed by senescence markers

Several functions of Tax are associated with genomic instability, repression of DNA damage response and induction of senescence [53]. Our results indicated significant deregulation of *TP53*, which differed between two infected clones (Fig 5). This trajectory resembled that of the divergent cell-cycle mediators in Fig 4B. By contrast, we observed up-regulation of another p53 family member (*TP63*) and many known p53 targets: *GADD45B*, *GADD45A*, *GADD45G*, *CDKN1A* (*p21*), and the main DNA-damage sensor of global genome nucleotide excision repair (GG-NER) *XPC*. We also observed up-regulation of *CETN2* and *RAD23B*, which together with *XPC* form the recognition complex of GG-NER [54]. Previously it has been reported that NER is suppressed by the direct up-regulation of *PCNA* induced by Tax [55,56]. *PCNA* was deregulated in both clones; however, the expression trajectories differed between the clones. The trajectories of two kinases activated by double-strand DNA breaks, *ATM* and *ATR*, differed in their response to plus-strand expression. *ATM* was significantly down-regulated in both clones, whereas *ATR* was up-regulated in clone 3.60 with a similar trend in TBX4B.

*CDKN1A*, which displayed a delayed up-regulation (Fig 5), belongs to the CIP/KIP family of CDK inhibitors. The CIP/KIP family gene *CDKN1C* (p57) was significantly up-regulated during the early transcriptional burst, with a higher level of expression in 3.60. The INK4 family of CDK inhibitor members, *CDKN2A* (p16) and *CDKN2B* (p15) were significantly down-regulated in both clones during proviral expression. In addition to *CDKN1A*, another senescence marker *GLB1* was up-regulated, peaking during mid-burst. These results indicate that DNA damage response pathways were activated during the burst, and the initial mitogenic signalling in G1 (Fig 4) was accompanied by the parallel down-regulation of G1-phase CDK inhibitors.

## IV. Proviral expression coincides with up-regulation of anti-apoptotic mediators and down-regulation of apoptotic effectors

Previous results have shown that Tax expression can promote cell death [23–27]. However, Tax has also been described to suppress apoptosis and this protection is transferable to cells not actively expressing Tax [15,16,20]. We observed a strong deregulation of key genes involved in the intrinsic and extrinsic apoptosis pathways during the plus-strand burst [57].

At the onset of the plus-strand burst, there was immediate downregulation of at least one of the pore-forming apoptotic factor genes in each clone (Fig 6). *BAX* was significantly down-regulated during the early-burst in clone 3.60, whereas a similar (yet non-significant) trend was seen in TBX4B. However, *BAK1* expression was sharply down-regulated only in clone TBX4B. The expression level of *BAK1* in clone 3.60 remained low during both the silent and early burst phases, but rebounded in the mid-burst and late burst.

Both clones showed strong up-regulation of the anti-apoptotic genes *BCL2*, *BCL2L1* and *BCL2L2*, which encode inhibitors of the pore-forming BCL2 family proteins (Fig 6). However, the pro-apoptotic genes *PMAIP1 (NOXA)*, *BCL2L11 (BIM)* and *BMF* were also immediately up-regulated. The pro-apoptotic gene *BID* was significantly up-regulated during the early-

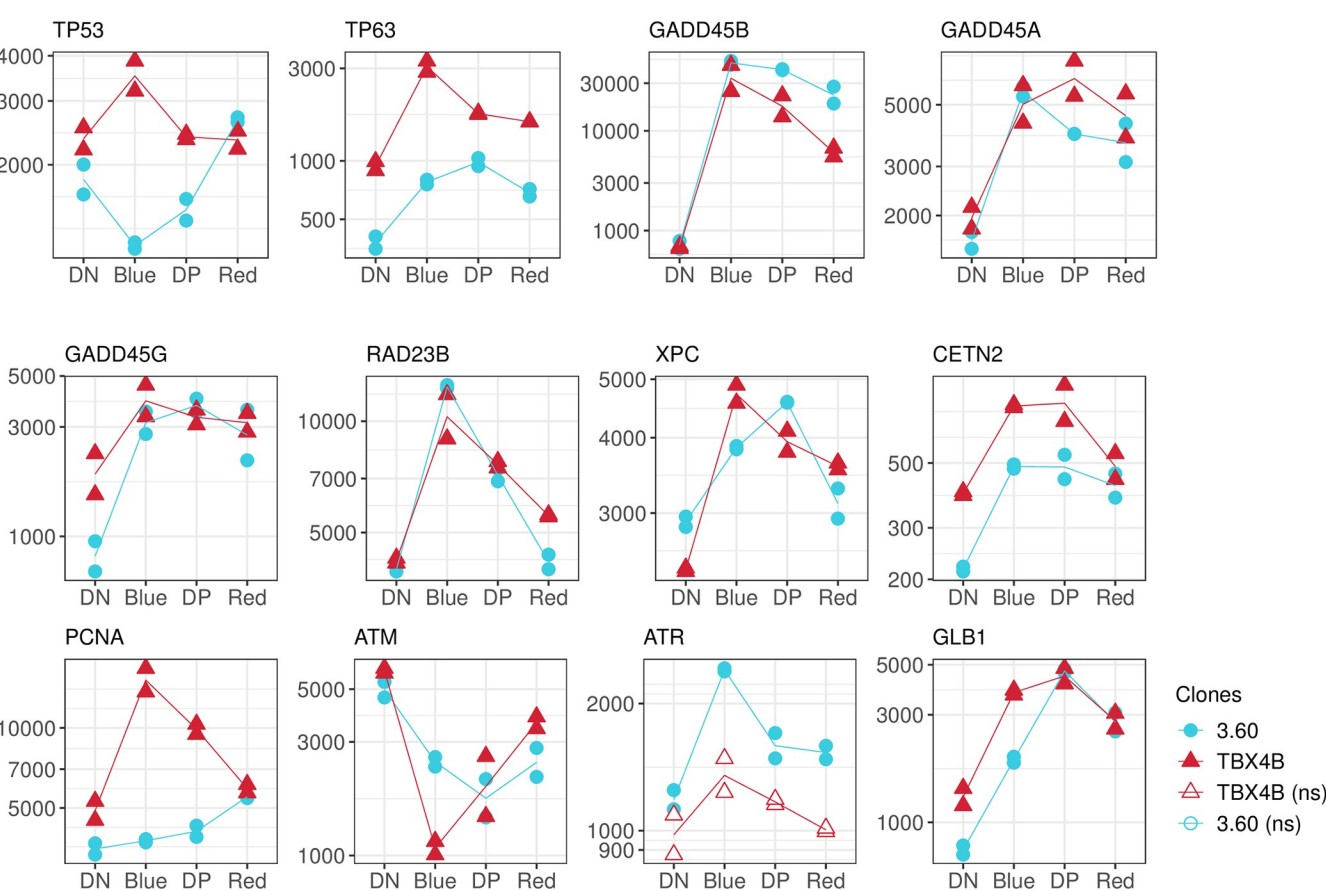

**Fig 5. Up-regulation of DNA damage response and senescence markers.** Gene expression trajectories of cell cycle inhibitors, DNA damage response genes and senescence markers. Y-axis: normalized counts on $\log_{10}$-scale. Statistical significance was determined by LRT. FDR-corrected p-value < 0.01; ns—not significant.

burst in clone 3.60. Curiously, both *PMAIP1* and *BCL2L11* sustained a high expression throughout proviral reactivation and remained high in the termination phase.

The death receptors *FAS* and *TNFRSF10B* were up-regulated in both clones and *TNFRSF10A* was up-regulated during the mid-burst in 3.60 yet down-regulated in clone

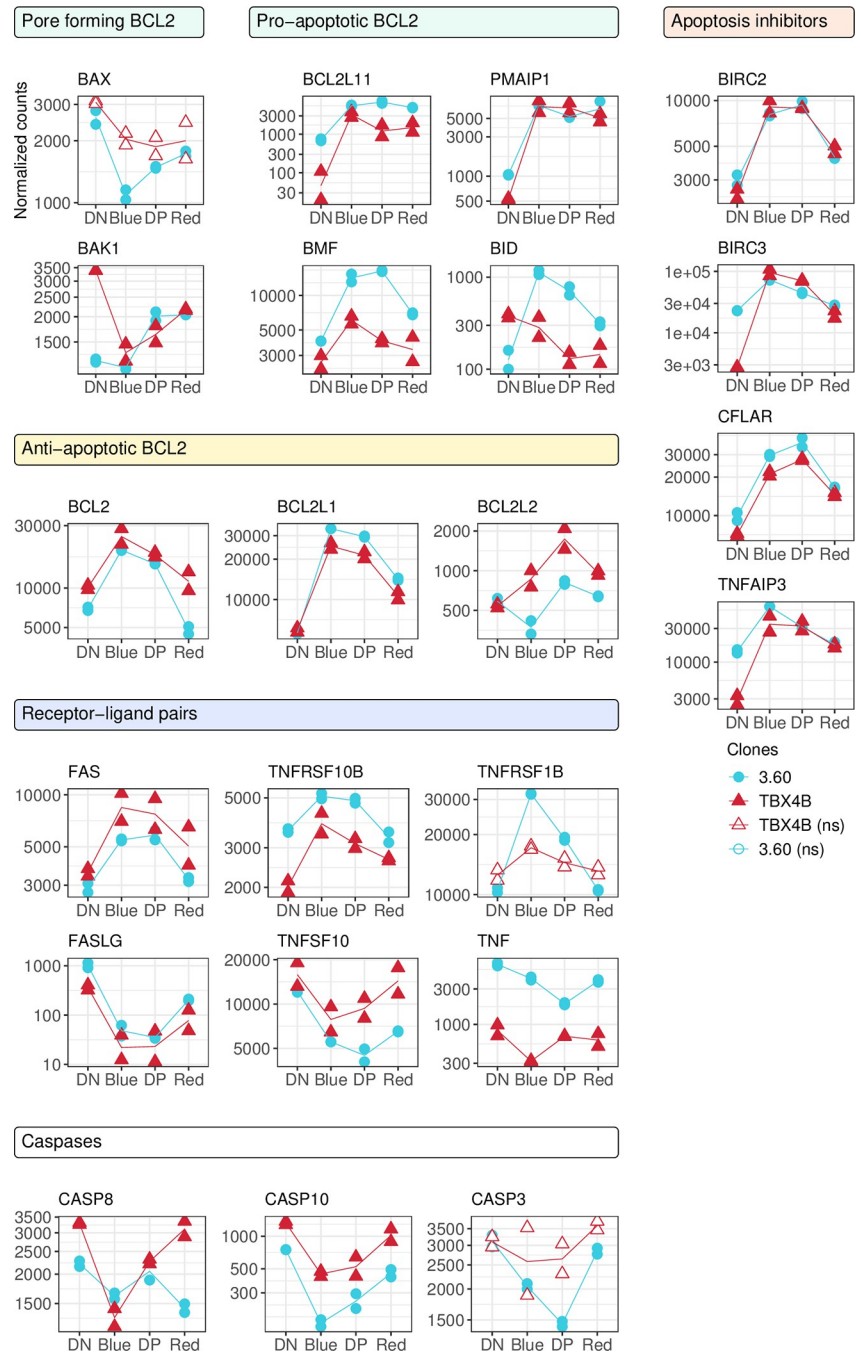

**Fig 6. Temporal patterns of pro- and anti-apoptotic factors.** Gene expression trajectories of anti- and pro-apoptotic BCL2 family members, extrinsic apoptosis factors, anti-apoptotic and caspase genes. Y-axis: normalized counts on $\log_{10}$-scale. Statistical significance was determined by LRT. FDR-corrected p-value < 0.01; ns—not significant.

TBX4B (Figs 6 and S3). Their cognate ligands *FASLG* and *TNFSF10* were strongly down-regulated.

Although the intrinsic pathway initiator *CASP9* was up-regulated only in clone 3.60, the death-inducing signaling complex (DISC) member genes *CASP8* and *CASP10*, which are initiators of the extrinsic pathway, were down-regulated during the early burst. The primary

effector *CASP3* was significantly downregulated over the course of proviral expression in clone 3.60, with a similar (albeit non-significant) trajectory in clone TBX4B. The inhibitors of apoptosis proteins capable of impairing caspase-mediated apoptosis—*BIRC2*, *BIRC3* and *CFLAR* (c-FLIP)—were strongly up-regulated in both clones.

A significant down-regulation of the granzyme genes *GZMA* and *GZMB* was also observed (S3 Fig). These genes are associated with cytotoxic activity of CD8[+] T-cells and NK cells; their function in CD4[+] T cells is incompletely understood.

These results showed that during the plus-strand burst the principal apoptotic effectors were down-regulated, and the apoptosis inhibitors up-regulated, in both extrinsic and intrinsic pathways. By contrast, a sustained expression of pro-apoptotic factor genes *PMAIP1* and *BCL2L11* outlasted the proviral burst.

## V. Increased expression of non-canonical polycomb repressive complex 1 members coincides with the plus-strand burst

The factors that regulate the spontaneous onset of expression of the provirus are not fully understood, but include the proviral integration site [58], cell stress [20,59], AhR signaling [60], and ubiquitinylation of histone 2A lysine 119 by polycomb repressive complex 1 (PRC1) [61].

*RING1*, *RYBP* and *KDM2B* are members of the non-canonical PRC1 (ncPRC1) [62] and their expression was up-regulated during the burst (Fig 7). *BMI1* (PCGF4), which is a core member of the canonical PRC1, was down-regulated in both clones (Fig 7).

## VI. Aryl hydrocarbon receptor (AhR) signaling augments HTLV-1 plus-strand expression, but not reactivation

We observed a consistent and robust differential expression of cytochrome P450 1A1 (*CYP1A1*), a product of the AhR pathway, between the early burst and late burst populations in both clones (S2 and S3 Figs). AhR is a transcription factor that regulates many biological processes through its activation in response to metabolic and environmental signals [63,64]. Following recent reports indicating enhanced HIV-1 proviral expression in response to AhR ligands in PBMCs isolated from patients on antiretroviral therapy [65] and HTLV-1 plus-strand expression in HTLV-1 infected transformed cell lines [60], we investigated the effect of AhR signalling on HTLV-1 expression in T-cell clones isolated from HTLV-1-infected individuals.

Endogenous AhR ligands such as tryptophan metabolites are present in the culture medium. We evaluated the effects of treatment with supplemental AhR ligands or AhR

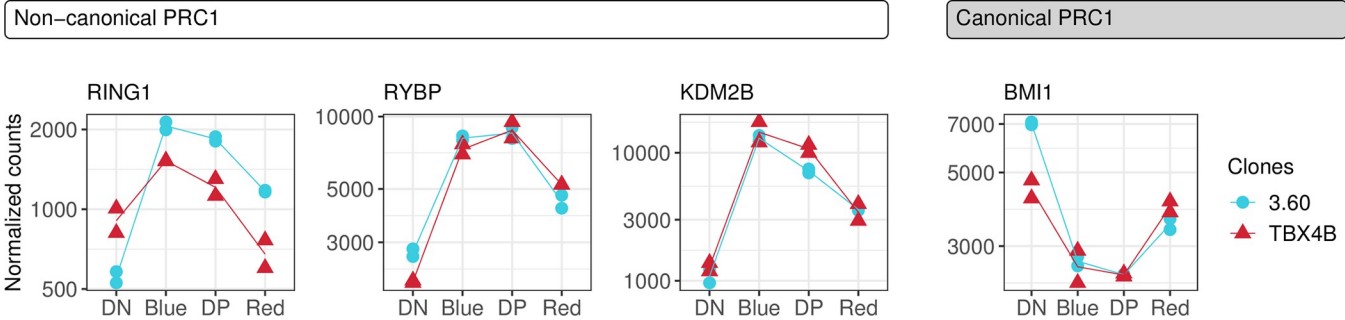

**Fig 7. Up-regulation of ncPRC1 members.** Gene expression trajectories of ncPRC1 and canonical PRC1 complex members. Y-axis: normalized counts on log$_{10}$-scale. Statistical significance was determined by LRT. FDR-corrected p-value < 0.01.

antagonists on HTLV-1 proviral expression using two patient-derived T-cell clones (3.60 and 11.50). Treatment with an endogenous AhR ligand, ITE [66], or a tryptophan-derived AhR ligand, FICZ [67] significantly increased Tax protein expression above background levels (Fig 8A). A purine-derived AhR antagonist StemRegenin 1 (SR1) [68] and a ligand-selective antagonist CH223191 [69] each substantially decreased Tax protein expression (Fig 8A). Transcription of the HTLV-1 plus-strand (*tax*) (Fig 8B) and AhR target genes (*CYP1A1* and *CYP1B1*) (Fig 8D and 8E) was significantly induced by AhR agonists and suppressed by AhR antagonists. Neither AhR agonists nor antagonists altered the expression of the HTLV-1 minus-strand (*sHBZ*) (Fig 8C).

The significant upregulation of *CYP1A1* expression observed during the late burst raised the question whether CYP1A1 itself contributes to the termination of HTLV-1 plus-strand expression (S1 and S2 Data, S2 and S3 Figs). However, treatment of the cells with Khellinoflavanone 4l (IIIM-517), an inhibitor of CYP1A1 enzymatic activity [70], did not affect HTLV-1 plus-strand expression (Fig 8A and 8B). We conclude that the observed up-regulation of *CYP1A1* indicated activation of the AhR pathway, but CYP1A1 itself is not directly involved in the termination of the HTLV-1 plus-strand burst.

We then investigated the effect of additional AhR ligands or inhibitors on HTLV-1 plus-strand reactivation and silencing dynamics using a patient-derived HTLV-1 infected T-cell clone (11.50) stably transduced with a Tax reporter construct that expresses a modified EGFP with a half-life of ~2h (d2EGFP). In these cells, the presence of d2EGFP is a surrogate for Tax protein expression. Live-cell imaging revealed that, compared with untreated cells, a greater portion of provirus-expressing cells terminated Tax expression in response to treatment with AhR antagonists (Fig 8F). Treatment with AhR agonists or a CYP1A1 inhibitor did not substantially affect proviral silencing or reactivation kinetics (Fig 8F and 8G). Spontaneous proviral reactivation was evident at early stages despite the presence of AhR inhibitors (Fig 8G).

These results indicate that enhanced AhR signalling augments and prolongs HTLV-1 plus-strand expression but is not the sole determinant of reactivation from latency in patient-derived T-cell clones.

## Discussion

It is well established that the HTLV-1 viral transactivator Tax deregulates the transcription of many host genes. Both Tax and the minus-strand-encoded HTLV-1 bZIP factor HBZ have been frequently implicated in leukemogenesis. *Tax* expression occurs in intermittent transcriptional bursts [20,38], likely in order to limit exposure to the immune system and the cytotoxic effects of Tax protein. It remains unclear whether the impact of HTLV-1 on host transcription, including genes involved in proliferation and apoptosis are immediate and direct, or late and indirect consequences of proviral reactivation and plus-strand expression. In this study, a Tax-responsive Timer Protein construct was used to distinguish successive temporal phases of the spontaneous proviral transcriptional burst, to investigate the precise trajectory of expression of host genes involved in cell cycle regulation and apoptosis during the proviral plus-strand burst.

Two naturally-infected T-cell clones competent in the expression of Tax allowed us to identify both clone-independent and clone-dependent correlates of proviral plus-strand expression. Clone 3.60 has a deletion in the coding sequence of *env* on the plus-strand and in the 3′UTR of the minus-strand-encoded gene *HBZ* (S1 Fig). This deletion does not change the predicted protein sequence of HBZ, and the deleted sequence was absent from the HBZ expression construct used to investigate the protein-dependent and mRNA-dependent actions

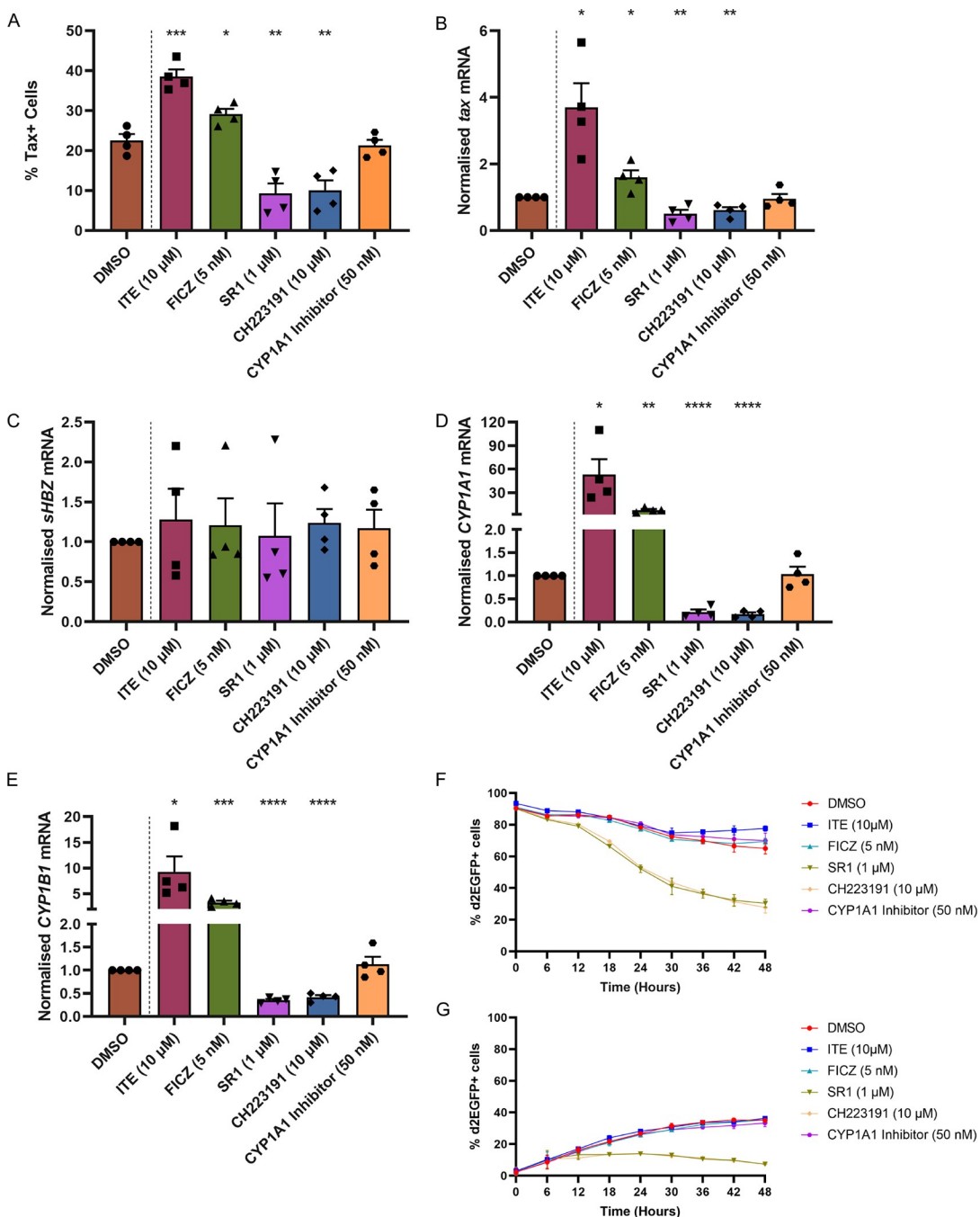

**Fig 8. AhR signalling enhances HTLV-1 plus-strand expression.** (A) Two patient-derived HTLV-1 infected T -cell clones were treated with AhR agonists or antagonists or CYP1A1 inhibitor for 48 hours at the indicated concentrations. DMSO was used as the vehicle control. The percentage of plus-strand expressing cells among viable cells was quantified by Tax protein expression using flow cytometry. The bar plot depicts the mean and SEM from two independent experiments. Unpaired two-tailed t-tests were used to determine the significance of the difference between the vehicle control and the treatment conditions. * $P < 0.05$, ** $P < 0.01$, *** $P < 0.001$. Expression levels of (B) *tax* (plus-strand), (C) *sHBZ* (minus-strand), AhR target genes (D) *CYP1A1* and (E) *CYP1B1* quantified by RT-qPCR after 24-hour treatment with DMSO, AhR activators or inhibitors or CYP1A1 inhibitor. Bar plots represent the mean and SEM from two independent experiments using two T-cell clones. * $P < 0.05$, ** $P < 0.01$, *** $P < 0.001$, **** $P < 0.0001$ (unpaired two-tailed t-test). Proviral (F) silencing and (G) reactivation kinetics in response to treatment with AhR agonists and antagonists. The data depict mean ± SEM from two independent experiments using a single clone. Individual trajectories are shown in S5 Fig.

of *HBZ* [71]. However, it remains possible that this deletion influences the half-life or the physiological actions of *HBZ* mRNA.

Proviral expression deregulated a large number of host genes in each clone: 3851 genes were deregulated in both clones, which could be grouped into clusters defined by the trajectory of expression during the proviral plus-strand burst (Fig 3A). Some of these observed effects may be due to other plus-strand encoded products; however, the cells were sorted on the expression of the Timer protein, which is directly induced by Tax. Tax activates both the canonical and non-canonical NF-κB pathways [72]. NF-κB pathway is persistently activated at the population level in transformed cell lines and primary ATL cells [73,74]. The present results confirm immediate, clone-independent NF-κB activation during proviral plus-strand expression (Figs 2A and 3 cluster 1), which is followed by the likely secondary effects of NF-κB activation (Fig 3 cluster 2); both the immediate and later effects decreased during the termination of proviral expression (Fig 3 clusters 1–2). *TNF* itself was downregulated (S1 and S2 Data and Fig 6): the observed up-regulation of genes in this enrichment term are likely to be the consequences of NF-κB activation by Tax. These observations suggest that high levels of NF-κB activation are confined to the active expression of Tax protein accompanying the plus-strand burst; constitutive activation of NF-κB may not be required for persistence in non-malignant HTLV-1-infected clones. We also confirmed the immediate up-regulation of *IL2RA* (Fig 2A), a known target of Tax, and genes in the "IL2-STAT5 signaling" enrichment term (Fig 3 cluster 1).

There was no consistent relationship between plus-strand and minus-strand expression of the provirus within each clone or between the clones (Fig 2A). The expression trajectory of the minus-strand resembled that of *SP1* (S3 Fig), a known regulator of *HBZ* expression [45]. These results suggest that it is unlikely that Tax directly regulates the expression of HBZ or vice versa, and imply that the clone-independent responses to HTLV-1 proviral reactivation observed in this study are not regulated by HBZ during the plus-strand burst.

Tax is known to up-regulate the expression of several genes involved in cell cycle progression including *CCND1*, *CCND2*, *CDK4*, *CDK6*, *CDK2* and *E2F1* [75–80]. The present results demonstrate a clone-specific association between proviral plus-strand expression and the expression of genes involved in cell cycle regulation (Fig 3 cluster 4). There was a strong difference between the two clones in the expression of genes that peak in different cell cycle phases obtained from Cyclebase 3.0 database (Fig 4A). Increased expression of G1-phase cyclins *CCND1*, *CCND2* and kinase *CDK6* suggests that the cells are stimulated to enter the cell cycle in each clone. In line with this, G1-phase CDK inhibitors *CDKN2A* and *CDKN2B* were downregulated (Fig 6). However, subsequent expression of G1/S, S and M phase genes *CCNE2*, *CCNA1*, *CCNA2*, *CCNB1*, *CCNB2* and *CDK1*, and key transcription factors that regulate the cell cycle, *E2F1* and *FOXM1*, diverged between the clones progressively throughout the cell cycle (Fig 4B). These findings are consistent with previous observations on the same clones reported by [38]. Billman et al. showed that Tax-expressing cells were more abundant in G1 phase in clone 3.60 and in G2/M phase in clone TBX4B. We note that clone 3.60 also grows more slowly in cell culture. Although *E2F1* is up-regulated in response to Tax [76,77], the diverging trajectories of *E2F1* and its downstream targets indicate that the transcription of *E2F1* is unlikely to be directly activated by Tax (Fig 4B). These observations emphasize the natural heterogeneity of HTLV-1 infected T-cells and help to reconcile previously published diverging results on cell-cycle progression in HTLV-1-infected cells.

It has been proposed that the interplay between the effects of Tax in proliferation and the DNA damage response regulates the fate of Tax-expressing cells [53]. Tax expression causes double-strand breaks and activation of the DNA damage response [5,29]; however, in the presence of additional genotoxic agents these pathways are impaired [53]. The activity of p53

is also repressed in HTLV-1-infected cells, through mechanisms that do not involve its DNA-binding activity and intracellular localization [81]. Our results show that *TP53* itself is deregulated during the plus-strand burst: the trajectory differed between the clones, but expression returned to baseline in each clone after termination of the burst (Fig 5). There was consistent up-regulation of another p53 family gene, *TP63*, and p53 targets including the GADD45 family members, *CDKN1A* and *XPC* (Fig 5). There was immediate up-regulation of genes involved in DNA damage response during the early burst of proviral expression, including *GADD45B*, *ATR* and global genome nucleotide excision repair (GG-NER) genes *RAD23B*, *XPC* and *CETN2* (Fig 5). However, *ATM* was down-regulated. ATR is known to respond to a wide range of DNA damage; the observed differences in expression between *ATR* and *ATM* in these clones indicates the presence of DNA damage other than double-stranded breaks. The up-regulation of DNA damage response genes was followed by the up-regulation of senescence markers *CDKN1A* and *GLB1*, which peaked during the mid-burst phase (Fig 5). Up-regulation of *CDKN1A* (p21) associated with hyperactivation of NF-κB by Tax has been shown to cause cell senescence [14]; however, the present results indicate that the up-regulation of *CDKN1A* and *GLB1* occurs in the mid-burst phase of persistent Tax expression, yet reduces during the late phase of the burst. The results demonstrate temporal separation of the DNA damage response and up-regulation of senescence-related genes during the plus-strand burst, and that sustained proviral expression may result in reduced proliferative capacity of HTLV-1 infected cells.

Tax has been shown to deregulate both pro-apoptotic and anti-apoptotic genes [17–22]. Consistent with this, both clones strongly up-regulated anti-apoptotic genes *BCL2*, *BCL2L1*, *BCL2L2*, *BIRC2*, *BIRC3*, *CFLAR* and *TNFAIP3*; and down-regulated key effectors of both the extrinsic and intrinsic apoptosis pathways: either *BAX or BAK1*, and *CASP3*, and down-regulated death receptor ligands *FASLG*, *TNFSF10*, *TNF* (Fig 6). Although the death receptor ligands were down-regulated, the death receptors *FAS* and *TNFRSF10B* were up-regulated in both clones (Fig 6). These observations suggest that the deregulation of genes involved in the extrinsic and intrinsic pathways is an immediate—perhaps direct—effect of Tax. Sustained expression of Tax is toxic to cells, and the up-regulation of pro-apoptotic factors can partly explain this; however, the results presented here suggest that the strong up-regulation of anti-apoptotic factors can counteract the pro-apoptotic effect during proviral plus-strand expression. Curiously, we observed an up-regulation of pro-apoptotic factors *PMAIP1* and *BCL2L11* that was sustained throughout the transcriptional burst and termination phases, which may pose an increased risk of apoptosis after termination of the plus-strand burst.

HTLV-1 proviral latency is associated with the PRC1-mediated ubiquitylation of histone 2A lysine 119 (H2AK119ub1); inhibition of deubiquitylation represses proviral plus-strand reactivation [61]. Here, we observed the up-regulation of ncPRC1 members *RING1*, *RYBP* and *KDM2B* through the early and mid-burst phases of proviral expression (Fig 7). Conversely, a core component of the canonical PRC1, *BMI1* (PCGF4), was down-regulated. Targeted recruitment of PRC1 to non-methylated CpG islands is mediated by KDM2B [82], and RYBP elevates the enzymatic ability of the PRC1 complex resulting in enhanced deposition of the H2AK119ub1 mark [83]. The up-regulation of these key PRC1 genes during proviral reactivation could be involved in the post-burst repression of proviral expression.

Up-regulation of genes involved in the inflammatory response and hypoxia (Fig 3, cluster 1) is consistent with the observation that cellular stress, including hypoxia, enhances proviral expression [20,59]. Although the hypoxia response observed by Kulkarni et al. was HIF-1-independent, we found that *HIF1A* was significantly up-regulated (S3 Fig).

Constitutive high expression of AhR, which is up-regulated in response to Tax, has been observed in ATL cells [84]. Recently, it was shown that persistent activation of NF-κB is important for the observed AHR expression: AhR signaling sustains and drives HTLV-1 plus-

strand expression and can potentiate HTLV-1 reactivation from latency [60]. Consistent with a previous report [84], we saw expression of *AHR*, *ARNT* and direct targets of AhR activation (*CYP1B1*, *NQO1*) in the silent population of cells in both clones. By contrast, proviral plus-strand expression was not associated with increased expression of *AHR*, but instead was accompanied by the down-regulation of *AHR*, *ARNT* and down-stream genes (S3 Fig). The effects observed here of treatment with agonists or antagonists of AhR (Fig 8) and the down-regulation of genes involved in the AhR pathway during spontaneous HTLV-1 proviral reactivation (S3 Fig), suggest that AhR activation enhances and prolongs proviral plus-strand expression, but *AHR* is transcriptionally inhibited during spontaneous HTLV-1 plus-strand expression. The transcriptional inhibition of the AhR pathway during the proviral burst may limit the extent and duration of Tax expression.

HTLV-1 expression, and in particular the Tax protein, have been associated with many transcriptional changes in the infected host T cell. The results presented here make it possible to distinguish between the immediate effects of the HTLV-1 plus-strand burst on host transcription, and the delayed or secondary effects. These results also demonstrate both clone-dependent and clone-independent transcriptional responses of the host cell accompanying the proviral plus-strand transcription. NF-κB was activated in response to HTLV-1 reactivation and this activation was contained to the duration of the proviral burst, which suggests NF-κB-mediated effects are not persistently active in clonal populations of naturally-infected T-cells. The regulation of genes responsible for progression through the cell cycle was clone-specific, emphasising the heterogeneity of naturally HTLV-1-infected T-cells. However, the up-regulation of genes involved in DNA damage recognition (GG-NER) and senescence were clone-independent, and associated with active expression of the provirus. Similarly, the transcriptional control of pro- and anti-apoptotic genes was consistent in the two clones and suggested a strong anti-apoptotic response that is limited to the duration of the burst; upregulation of certain pro-apoptotic genes outlasted the burst. We also observed the up-regulation of non-canonical PRC1 members, which are associated with the epigenetic regulation of the provirus [61]. In the context of these results, it will be important to verify the effects of this transcriptional regulation of host genes, on the dynamics of infected cells during and following the burst. Finally, we tested the involvement of the AhR pathway in proviral reactivation and found that while activation of the AhR pathway increased the intensity of the plus-strand burst, it did not increase the frequency of reactivation.

## Materials and methods

### Cell culture

The HTLV-1-infected clones used in this study were CD4$^+$CD25$^+$CCR4$^+$ T cells, each carrying a single copy of the HTLV-1 provirus, derived from peripheral blood cells isolated from HTLV-1-infected individuals as described previously [40]. The clones were cultured in RPMI-1640 (Sigma-Aldrich) supplemented with 20% fetal bovine serum (FBS), 2 mM L-Glutamine, 50 IU/ml Penicillin, 50 μg/ml Streptomycin (all from ThermoFisher Scientific) and 100 IU/ml human interleukin 2 (IL-2, Miltenyi Biotec). Ten micromolar integrase inhibitor, Raltegravir (Selleck Chemicals) was added to the cultures to prevent secondary HTLV-1 infections. The cells were supplemented with IL-2 and Raltegravir twice-weekly intervals and cultured at 37˚C, 5% $CO_2$.

### Plasmid generation

To create pLJM1-LTR-FT, pLJM1-EGFP (Addgene 19319) was digested with NdeI and EcoRl to create the vector backbone. A forward primer (5'-ATGGTGAGCAAGGGCGAG-3') and a

reverse primer (5'-TCGAGGTCGAGAATTCTTACTTGTACAGCTCGTCCATGC-3') with a 15 base pair overlap with vector backbone were used in a polymerase chain reaction (PCR) to generate fast Timer protein timer fragment from plasmid pFast-FT-N1 (Addgene 31910). Five tandem repeats of Tax responsive element (TRE) type 2 and an HTLV-1 promoter was amplified from WT-Luc plasmid [85] by PCR using forward (5'-AAATGGACTATCATATGGGG AGGTACCGAGCTCTTACGC-3') and reverse (5'-GCCCTTGCTCACCATGGTGGCGGGC CAAGCCGGCAGTCA-3') primers with 15 base pair overlap with vector backbone and fast timer protein PCR product, respectively. Two PCR products were inserted into the vector backbone using In-Fusion HD Cloning Kit (Takara Bio) to generate pLJM1-LTR-FT. The sequence of the inserts was verified by Sanger sequencing (GATC Biotech).

pLJM1-LTR-d2EGFP was generated by digesting pLJM1-EGFP with Ndel and EcoRl to produce the vector backbone. A PCR incorporating a forward primer (5'-GCCACCATGGT GAGCAAGG-3') and a reverse primer (5'-TCGAGGTCGAGAATTCCTACACATTGATCC TAGCAGAAGC-3') with a 15 base pair overlap with vector backbone were used to amplify destabilised enhanced green fluorescent (d2EGFP) fragment from pcDNA3.3_d2eGFP plasmid (Addgene 26821). A fragment containing 9 copies of TRE type 1 and TRE type 3 and an HTLV-1 promoter was amplified from SMPU-18x21-EGFP plasmid [86] by PCR using forward (5'-AAATGGACTATCATATGCGGGTTTATTACAGGGACAGCG-3') and reverse (5'-GCTCACCATGGTGGCATCTCGCCAAGCTTGGATCTGT-3') primers with 15 base pair overlap with vector backbone and d2EGFP PCR product, respectively. pLJM1-LTR-d2EGFP was formed by inserting the two PCR products into the vector backbone using In-Fusion HD Cloning Kit. Sanger sequencing was used to verify the sequence of the inserts in the transfer plasmid.

## Lentiviral transduction

HEK 293T cells were seeded into 150 mm Corning TC-treated Culture Dishes (Corning) the day before transfection to reach an approximately 95% confluence on the day of transfection. HEK 293 T cells were co-transfected with either pLJM1-LTR-FT or pLJM1-LTR-d2EGFP, psPAX2 (Addgene 12260) and pCMV-VSV-G (Addgene 8454) plasmids using Lipofectamine 3000 (Invitrogen) following the manufacturer's protocol. Viral supernatants were harvested 24 and 52 hours post-transfection. Supernatants were centrifuged at 2000 rpm for 10 minutes and passed through a 0.45 μm syringe filter (Sartorius) to remove debris prior to concentration by ultracentrifugation at 25000 rpm for 2 hours at 4°C. One hundred thousand cells were spinoculated with 100 μl of concentrated viral supernatant in the presence of 8 μg/ml polybrene and 10 mM HEPES (Sigma-Aldrich) at 800 g, 32°C for 2 hours. Transduced cells were washed once and cultured in complete medium supplemented with IL-2. Three days post-transduction, the cultures were supplemented with Raltegravir and Puromycin Dihydrochloride (ThermoFisher Scientific) was added at 2 μg/ml twice a week for 14 days to select transduced cells. Timer protein or d2EGFP-positive cells were sorted by flow cytometry to obtain Timer protein or d2EGFP-expressing populations. Flow-sorted cultures returned to their steady state within two weeks of flow sorting. Replicates were grown in parallel. The cultures were maintained in 1 μg/ml Puromycin Dihydrochloride during the regular feeding cycle to prevent the emergence of resistance gene silent populations. The transduced clones expressing two different Tax reporter systems are given in Table 1.

## Flow cytometry analysis

Cells were washed once with PBS and stained with 1 μg/ml viability marker, LIVE/DEAD fixable near-IR (ThermoFisher Scientific) for 5 minutes and washed once in FACS buffer (PBS

**Table 1. Patient-derived T-cell clones expressing fluorescent protein-based Tax reporter systems.**

| Clone | Integration site (GRCh38) | Tax reporter system |
|---|---|---|
| TBX4B | Chr 22:43927318 | pLJM1-LTR-FT |
| TBJ 3.60 | Chr 4:69701567 | pLJM1-LTR-FT |
| TBW 11.50 | Chr 19:27791679 | pLJM1-LTR-d2EGFP |

+ 5% FCS), and fixed for 30 minutes with fixation/permeabilization buffer of eBioscience FOXP3/Transcription Factor Staining Buffer Set (ThermoFisher Scientific). The cells were then washed with permeabilization buffer, stained with 1 µg/ml anti-Tax mAb (Clone LT-4) in permeabilization buffer for 30 minutes, washed twice in permeabilization buffer, and resuspended in FACS buffer. A slightly modified staining protocol was used to co-detect Tax protein with timer protein. Following the staining with viability marker and subsequent wash, the cells were fixed with 4% formaldehyde (ThermoFisher Scientific) for 15 minutes, washed once in FACS buffer, permeabilized with 0.1% Triton X-100 (ThermoFisher Scientific) for 15 minutes, washed once with FACS buffer and stained with 1 µg/ml anti-Tax mAb (Clone LT-4) in FACS buffer for 30 minutes. Finally, the cells were washed twice and resuspended in FACS buffer. All washes and incubations were performed at room temperature for flow cytometry analysis and sorting. The cells were acquired on a BD LSRFortessa (BD Biosciences) flow cytometer. FlowJo software (BD Biosciences) was used to analyse flow cytometry data.

## Flow sorting

Live cell flow cytometry sorting under containment level 3 (CL3) conditions was performed in the CL3 Cell Sorting Facility at Chelsea and Westminster Hospital in London. Cells were washed once with PBS and stained with 1 µg/ml LIVE/DEAD fixable near-IR viability dye for 5 minutes, washed once and resuspended in RPMI 1640 without phenol red (ThermoFisher Scientific) supplemented with 2% FCS. Viable Blue$^-$Red$^-$ (double negative, DN), Blue$^+$Red$^-$, Blue$^+$Red$^+$ (double positive, DP), Blue$^-$Red$^+$ or viable d2EGFP$^+$ and d2EGFP$^-$ cells were sorted under sterile conditions using a BD FACSAria lll cell sorter. Duplicate parallel cultures from each timer protein Tax reporter clone were flow-sorted on the same day. RNeasy Plus Micro Kit (Qiagen) was used to extract RNA from the flow-sorted timer protein sub-populations following the manufacturer's protocol. RNA integrity was quantified using RNA 6000 Pico Kit (Agilent) on a 2100 Bioanalyzer (Agilent).

## Live-cell imaging

Flow-sorted proviral-expressing (d2EGFP$^+$) and non-expressing (d2EGFP$^-$) cells were seeded into a 96 well plate pre-coated with 1 mg/ml Poly-D-Lysine (PDL, Merck). Aryl hydrocarbon receptor (AhR) agonists and antagonists were added at concentrations indicated in Fig 8. One hundred nanomolar YOYO-3 Iodide (ThermoFisher Scientific) was added to label dead cells. Live-cell imaging was performed using Incucyte S3 (Sartorius) live-cell imaging system capturing 9 Phase contrast, green and red fluorescent images per well every 6 hours using a 20x objective. Image analysis was performed with the "Non-adherent Cell-by-Cell" image analysis module on the Incucyte, using the parameters listed in Table 2. The percentage of viable cells that were d2EGFP positive was calculated.

## Quantitative real-time PCR

RNeasy Plus Mini kit (Qiagen) was used to extract RNA from cells cultured with vehicle control (DMSO), or AhR agonists or antagonists. RNA was reverse-transcribed using Transcriptor

**Table 2. Imaging and mask parameters used for image capturing and analysis on Incucyte S3.**

| Channel | Target | Exposure time | Background fluorescence correction method | Segmentation parameters |
|---|---|---|---|---|
| Phase | All cells | Not available | Not applicable | Sensitivity (Threshold = 9, Background = 10, Edge = 10, Particle area (minimum = 30 μm$^2$, maximum = ∞ μm$^2$) |
| Green | d2EGFP + Cells | 300 ms | Top-Hat (50 μm radius) | Not applicable |
| Red | Dead cells | 400 ms | Top-Hat (50 μm radius) | Not applicable |

First Strand cDNA Synthesis Kit (Roche) with random hexamer primers following manufacturer's instructions. A no-reverse transcriptase (RT) control was included for each sample to verify the elimination of genomic DNA from RNA samples. RNA transcripts were amplified with a master mix containing gene-specific primers listed in Table 3 and Fast SYBR Green Master Mix (ThermoFisher Scientific) on a Viia 7 Real-Time PCR System (ThermoFisher Scientific). The relative quantification of target mRNAs was performed using the LinRegPCR method [87], and the data were normalised against the internal PCR control, 18S rRNA.

## Statistical analysis

Statistical analysis was performed using GraphPad Prism (GraphPad Software) and in R [90].

## RNA-seq Alignment and quantification

Paired-end 150 bp poly-A enriched stranded RNA libraries were prepared with NEBNext Ultra II Directional RNA Library Prep Kit for Illumina. Reads were sequenced on the Illumina's HiSeq 4000 Sequencing System by Oxford Genomics Centre. Samples were sequenced in two lanes and the resulting FASTQ files aggregated for each sample. FastQC (RRID: SCR_014583, version 0.11.8) and MultiQC (RRID:SCR_014982, version 1.8) were used for quality assessment before and after adapter and quality trimming with Trim Galore (RRID: SCR_011847, version 0.6.4_dev). The STAR aligner (RRID:SCR_004463, version 2.7.3a) was used to align reads against a custom merged reference of the human (Ensembl100 GRCh38) genome [91], HTLV-1 (GenBank: AB513134) genome and the reference sequence of the Timer protein. A custom gene transfer format (GTF) including coordinates for the Timer protein and HTLV-1 was also supplied for STAR to transform the alignments into transcript coordinates (—quantMode TranscriptomeSAM). RSEM (RRID:SCR_013027, version 1.3.1) was then used for transcript quantification of stranded aligned reads (—forward-prob 0) [92].

**Table 3. Gene-specific primers used for RT-qPCR.**

| Target gene | Orientation | Sequence | Reference |
|---|---|---|---|
| *tax* | Forward | 5'-CCGGCGCTGCTCTCATCCCGGT-3' | [88] |
| | Reverse | 5'-GGCCGAACATAGTCCCCCAGAG-3' | |
| *sHBZ* | Forward | 5'-GGACGCAGTTCAGGAGGCAC-3' | |
| | Reverse | 5'-CCTCCAAGGATAATAGCCCG-3' | |
| *18S* | Forward | 5'-GTAACCCGTTGAACCCCATT-3' | |
| | Reverse | 5'-CCATCCAATCGGTAGTAGCG-3' | |
| *CYP1A1* | Forward | 5'-CACCATCCCCCACAGCAC-3' | [89] |
| | Reverse | 5'-ACAAAGACACAACGCCCCTT-3' | |
| *CYP1B1* | Forward | 5'-GCTGCAGTGGCTGCTCCT-3' | |
| | Reverse | 5'-CCCACGACCTGATCCAATTCT-3' | |

## Differential expression analysis

Tximport (RRID:SCR_016752, version 1.14.2) was used to import gene level transcript abundance estimates for differential expression analysis using DESeq2 (RRID:SCR_015687, version 1.32.0) in R (version 4.1.2) [90,93,94]. Each clone was analysed separately, and genes were included for analysis if detected at a minimum of 3 reads in 2 or more of the samples. The LRT was used to identify significantly DE genes across all Timer protein populations; for pairwise comparisons the default Wald test was used. FDR adjusted p-value < 0.01 cut-off was used for both approaches [95].

## K-means clustering

K-means clustering (using MacQueen algorithm) with k = 5 was carried out in R on the subset of DE genes that overlapped in the two clones [90]. A joined matrix of the scaled variance-stabilizing transformation (VST) transformed counts was used as input [96].

## Over-representation analysis

Over-representation analysis (ORA) was performed with g:Profiler (RRID:SCR_006809, version 0.2.0) against the MSigDB's Hallmark gene set collection [97,52]. A custom background of all genes that were subjected to differential expression testing in both clones was used.

## Supporting information

**S1 Fig. Schematic of provirus in 3.60.** Schematic of the provirus in clone 3.60 with the 202 bp deletion (GenBank: AB513134; coordinates 6420–6621) and coding-regions of Env, Tax and sHBZ marked.
(PDF)

**S2 Fig. Volcano plots of genes deregulated between early burst and silent and late burst phases.** Significantly up-regulated genes are in yellow and down-regulated genes in blue. HTLV-1 plus, Timer and top 10 most significantly up- and down-regulated genes are labelled; ns—not significant.
(PDF)

**S3 Fig. Gene trajectories of genes mentioned in main text.** Y-axis: normalized counts on log10-scale. Significance is determined with LRT. FDR-corrected p-value < 0.01.
(PDF)

**S4 Fig. NF-κB pathway is significantly up-regulated in each clone.** (A) K-means clustering of 10048 significantly DE genes with k = 2 in clone 3.60 and 4798 DE genes in clone TBX4B. The top 10 genes based on mean rank of sorted p-values are listed on the right in each panel. The mean expression trajectory is coloured yellow or blue. Significance is determined with LRT. FDR-corrected p-value < 0.01. (B) Over-representation analysis of K-means clusters with the Hallmarks gene set from The Molecular Signatures Database (MSigDB). Statistical significance was determined by Fisher's exact test in g:Profiler. FDR-corrected p-value < 0.05.
(PDF)

**S5 Fig.** (A) Tax expression analysis by flow cytometry. NT–non-transduced. (B) *tax* expression analysis RT-qPCR. (C) *sHBZ* expression analysis RT-qPCR. (D) *CYP1A1* expression analysis RT-qPCR. (E) *CYP1B1* expression analysis RT-qPCR.
(PDF)

**S1 Data. Differential expression results for clone 3.60.**
(XLSX)

**S2 Data. Differential expression results for clone TBX4B.**
(XLSX)

**S3 Data. AhR measurements.**
(ZIP)

## Acknowledgments

We thank Parisa Amjadi from the CL3 Cell Sorting Facility at The Centre for Immunology and Vaccinology at Imperial College London. We thank Oxford Genomics Centre for library preparation and RNA Sequencing. We thank Chou-Zen Giam (Uniformed Services University) for providing the SMPU-18x21-EGFP plasmid; Brigitta Stockinger (Crick Institute) for providing FICZ and helpful discussions; and Sandip Bharate (Indian Institute of Integrative Medicine) for providing CYP1A1 inhibitor, IIIM-517. We thank Imperial College Research Computing Service for use of the high-performance computing cluster (DOI: 10.14469/hpc/2232).

## Author Contributions

**Conceptualization:** Helen Kiik, Saumya Ramanayake, Michi Miura, Anat Melamed, Charles R. M. Bangham.

**Data curation:** Helen Kiik.

**Formal analysis:** Helen Kiik, Saumya Ramanayake.

**Funding acquisition:** Charles R. M. Bangham.

**Investigation:** Saumya Ramanayake.

**Methodology:** Helen Kiik, Saumya Ramanayake, Michi Miura.

**Project administration:** Charles R. M. Bangham.

**Resources:** Yuetsu Tanaka, Charles R. M. Bangham.

**Software:** Helen Kiik.

**Supervision:** Anat Melamed, Charles R. M. Bangham.

**Validation:** Helen Kiik, Saumya Ramanayake.

**Visualization:** Helen Kiik, Saumya Ramanayake.

**Writing – original draft:** Helen Kiik, Saumya Ramanayake, Charles R. M. Bangham.

**Writing – review & editing:** Helen Kiik, Saumya Ramanayake, Michi Miura, Yuetsu Tanaka, Anat Melamed, Charles R. M. Bangham.

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
