## [Decision Letter · Decision Letter 0]

21 Mar 2022

Dear Prof. Bangham,

Thank you very much for submitting your manuscript "Time-course of host cell transcription during the HTLV-1 transcriptional burst" for consideration at PLOS Pathogens. As with all papers reviewed by the journal, your manuscript was reviewed by members of the editorial board and by three independent reviewers. In light of the reviews (below this email), we would like to invite the resubmission of a significantly-revised version that takes into account the reviewers' comments.

In particular, 2 of the reviewers commented that the manuscript would be strengthened by some functional analyses of the cells, to determine if the expression studies can be confirmed biologically.

We cannot make any decision about publication until we have seen the revised manuscript and your response to the reviewers' comments. Your revised manuscript is also likely to be sent to reviewers for further evaluation.

Sincerely,

Susan R. Ross, PhD

Section Editor

PLOS Pathogens

Susan Ross

Section Editor

PLOS Pathogens

Kasturi Haldar

Editor-in-Chief

PLOS Pathogens

orcid.org/0000-0001-5065-158X

Michael Malim

Editor-in-Chief

PLOS Pathogens

orcid.org/0000-0002-7699-2064

Reviewer's Responses to Questions

**Part I - Summary**

Reviewer #1: In this manuscript, Bangham and colleagues examine transcriptional changes in viral and cellular gene expression in successive phases of the HTLV-1 plus-strand burst. Two HTLV-1-infected T cell clones (3.60 and TBX4B) were stably transduced with a Tax-responsive Timer protein which undergoes a blue-to-red conversion in fluorescence emission over time. The clones were flow sorted into four populations (Blue-Red-, Blue+Red-, Blue+Red+, Blue-Red+) and RNA sequencing was used to examine gene expression. These four populations represent silent proviruses (Blue-Red-), early phase (Blue+Red-), mid-phase (Blue+Red+) and late phase (Blue-Red+) of the plus-strand burst. As expected, there were distinct profiles of host gene expression in each of the phases with Tax-induced NF-kB target genes, including cytokines and apoptotic regulators, mainly expressed in the early phase of proviral reactivation. Downregulation of these genes in the late phase suggest that NF-kB activation is transient in these clones. Although both clones had similar patterns of gene expression for the majority of functional classes of genes, many cell cycle regulatory genes exhibited divergent expression in the clones. Finally, activation of the arylhydrocarbon receptor (AhR) pathway enhanced activation of HTLV-1 plus-strand expression. Overall, this is a well designed and interesting study that provides new insight into the dynamics of host cell gene expression after spontaneous proviral reactivation. It will be of great interest to the field and should serve as a valuable resource. The points below should be addressed by the authors to further substantiate the conclusions.

Reviewer #2: Kiik et al, in their manuscript titled “Time-course of host cell transcription during the HTLV-1 transcriptional burst” examined the host transcription profiles during the defined phases of the proviral plus strand transcriptional burst. Utilizing HTLV-1 infected T cell clones transduced with a tax responsive Timer protein which goes through a shift in fluorescent emissions over time, the authors were able to define 4 consecutive phases of the transcription burst; silent, early, mid and late. The authors showed that the expression profiles of genes in the NF-KB signaling pathway displayed similar trajectories, as that seen in the two clones during the HTLV-1 plus strand transcription. Conversely, the two clones showed divergence in the trajectories of genes controlling cell cycle. The authors also showed that there is an upregulation in a number of senescence and DNA damage Repair markers, most peaking during the early phase of the burst. A divergent pattern was observed in the pro-apoptotic genes and the anti-apoptotic genes. The pro-apoptotic genes are shown to be regulated in early burst, with continued expression through the late phase of the burst, suggesting cells may be prone to apoptosis at the conclusion of a burst. The authors showed that AhR signaling regulates the intensity and the duration of the plus strand transcriptional burst but not its reactivation from latency. This is a well-written manuscript describing the host transcription profiles during the defined phases of the proviral plus strand transcriptional burst.

Reviewer #3: In the present study Kiik et al. analyze the temporal changes in host transcription during successive phases of the HTLV-1 plus-strand burst in two naturally-infected T-cell clones isolated by limiting dilution from peripheral blood mononuclear cells (PBMCs) of HTLV-1-infected subjects

that were stably transduced with a Tax-responsive, time-sensitive reporter construct.

RNAseq was employed on color-sorted cells to investigate the changes in host cell transcription accompanying the HTLV-1 plus-strand bursts. Results showed a transient deregulation of genes involved in Tax-associated alterations, although the two clones diverged strongly in their expression of genes regulating the cell cycle. Activation of the arylhydrocarbon receptor (AhR) pathway enhanced and prolonged the proviral burst, but did not increase the rate of reactivation.

In general, this is an excellent study that applies state-of the-art techniques to tackle some key unanswered questions on the regulation of HTLV-1 gene expression and its connection with the pattern of host gene expression.

The experimental layout is sound and the data presented are solid.

**Part II – Major Issues: Key Experiments Required for Acceptance**

Reviewer #1: 1) The study is underpowered with only two HTLV-1-infected T cell clones, and one of the clones (3.60) has a deletion in the coding sequence of env and the 3’UTR (although it is unclear if this deletion has any effect on gene expression). Proviral integration sites can also influence gene expression in different clones. It is difficult to explain the divergent expression of cell cycle genes in the two clones since Tax upregulates many of these genes. Therefore, it is important to confirm key results and further investigate the expression of cell cycle genes by qRT-PCR in additional clones expressing the Tax-responsive Timer protein.

2) The inclusion of functional studies would strengthen the main conclusions of the study. The host gene expression profiles in each of the successive phases of the plus-strand burst suggest distinct activation states and susceptibility to cell death. Can the authors experimentally confirm functional changes between cells in different phases? For example, are cells in the late phase (Red) more susceptible to cell death upon activation of intrinsic or extrinsic cell death pathways?

Reviewer #2: Although the expression and trajectory of genes during the HTLV-1 plus strand/tax transcriptional burst is nicely defined specifically in two clones, it is necessary to highlight the viral gene expression in the clones. This will help clarify whether differences observed in expression/trajectory may be a result of the expression level of other viral genes.

Reviewer #3: none

**Part III – Minor Issues: Editorial and Data Presentation Modifications**

Reviewer #1: 1) What is the HTLV-1 integration site in clone 3.60?

2) More details are needed about the spontaneous proviral expression that occurs in the 3.60 and TBX4B clones. This group has previously demonstrated considerable heterogeneity in the bursts of plus-strand expression within the same clone (Billman et al. Wellcome Open Res. 2017). What is the length of the interval between the intermittent bursts of spontaneous proviral gene expression? Is this also heterogeneous at the single-cell level?

Reviewer #2: 1. The expression pattern of CDK2 discussed in line 223 was not present in the indicated figure (Fig4B) or in a supplementary figure.

2. The expression pattern of TNFRSF10A discussed in line 297 was not present in the indicated figure (Fig6) or in a supplementary figure.

3. The methodology for the dynamics/kinetics of tax expression to determine cell silencing and reactivation is not clearly defined in the methods provided in Figure 8 F, G. Were cells synchronized to be at the same phase at the initiation of the time course or treatment?

4. The use of 2 different T cell clones, whose viral gene expression are uncharacterized, as a mixed population to determine the effect of the AhR on proviral expression in Figure 8, makes it difficult to infer whether results are clone specific or can be generalized. Presentation of data for individual clones will help determine whether the observed regulation can be generalized.

5. Poor Image quality/ text compressed in Figure 3A, B and Figure 8F, G

Reviewer #3: Below are my comments/suggestions to improve the quality of this paper.

The sentence on lines 65-67 is oversimplistic: the plus strand encodes many other proteins, some of which have been directly implicated with regulation of gene expression (e.g. Rex, p30). It is thus possible that some of the observed effects are not due to Tax alone.

Was a geneset of CREB-responsive transcripts also enriched? This would be an expected result, especially as the expression from the HTLV-1 provirus (and Timer reporter) are CREB-driven. To this effect it was a bit surprising to find P/CAF downregulated (lines 161-162).

Lines 143-144 – The very large number of differentially expressed genes found (over 10,000 / 57% in one clone) raises some concerns on the biological relevance of these changes. Did the analysis pipeline include a cut-off for the number of reads and, more importantly, for the fold change? How many genes are differentially regulated if one considers a fold change of at least 2?

Do the divergent gene expression of cell cycle regulators correlate with different growth properties of the two clones?

It is surprising that NER-related genes were found to be involved, since oncogenic /mitogenic stress (and Tax) are more commonly associated with strand breaks and the ATM pathway.

PLOS authors have the option to publish the peer review history of their article (what does this mean?). If published, this will include your full peer review and any attached files.

Reviewer #1: No

Reviewer #2: No

Reviewer #3: No
---

## [Decision Letter · Decision Letter 1]

22 Apr 2022

Dear Prof. Bangham,

We are pleased to inform you that your manuscript 'Time-course of host cell transcription during the HTLV-1 transcriptional burst' has been provisionally accepted for publication in PLOS Pathogens.

Best regards,

Susan R. Ross, PhD

Section Editor

PLOS Pathogens

Susan Ross

Section Editor

PLOS Pathogens

Kasturi Haldar

Editor-in-Chief

PLOS Pathogens

orcid.org/0000-0001-5065-158X

Michael Malim

Editor-in-Chief

PLOS Pathogens

orcid.org/0000-0002-7699-2064

Reviewer Comments (if any, and for reference):

Reviewer's Responses to Questions

**Part I - Summary**

Reviewer #1: (No Response)

**Part II – Major Issues: Key Experiments Required for Acceptance**

Reviewer #1: No additional experiments were performed by the authors in the revised submission. However, I am satisfied with the authors' responses to my previous comments.

**Part III – Minor Issues: Editorial and Data Presentation Modifications**

Reviewer #1: (No Response)

PLOS authors have the option to publish the peer review history of their article (what does this mean?). If published, this will include your full peer review and any attached files.

Reviewer #1: No

---

## [Editor Report · Acceptance letter]

11 May 2022

Dear Prof. Bangham,

We are delighted to inform you that your manuscript, "Time-course of host cell transcription during the HTLV-1 transcriptional burst," has been formally accepted for publication in PLOS Pathogens.

Best regards,

Kasturi Haldar

Editor-in-Chief

PLOS Pathogens

orcid.org/0000-0001-5065-158X

Michael Malim

Editor-in-Chief

PLOS Pathogens

orcid.org/0000-0002-7699-2064